# Ribozyme-mediated RNA synthesis and replication in a model Hadean microenvironment

Annalena Salditt[1], Leonie Karr[1], Elia Salibi[2], Kristian Le Vay[2], Dieter Braun [1] ✉ & Hannes Mutschler [2] ✉

Enzyme-catalyzed replication of nucleic acid sequences is a prerequisite for the survival and evolution of biological entities. Before the advent of protein synthesis, genetic information was most likely stored in and replicated by RNA. However, experimental systems for sustained RNA-dependent RNA-replication are difficult to realise, in part due to the high thermodynamic stability of duplex products and the low chemical stability of catalytic RNAs. Using a derivative of a group I intron as a model for an RNA replicase, we show that heated air-water interfaces that are exposed to a plausible $CO_2$-rich atmosphere enable sense and antisense RNA replication as well as template-dependent synthesis and catalysis of a functional ribozyme in a one-pot reaction. Both reactions are driven by autonomous oscillations in salt concentrations and pH, resulting from precipitation of acidified dew droplets, which transiently destabilise RNA duplexes. Our results suggest that an abundant Hadean microenvironment may have promoted both replication and synthesis of functional RNAs.

Identifying a physicochemical environment suitable for auto-catalytic template-dependent self- and cross-replication cycles of RNAs would significantly advance our understanding of plausible scenarios that could explain the emergence of life on Earth[1–3]. The development of catalytic RNAs (ribozymes) capable of accelerating RNA ligation and polymerization over the last decades has tremendously progressed the field of protein-free RNA replication[4–8]. However, the high thermodynamic stability of the RNA duplexes resulting from these reactions, which is further enhanced by salts present in the solution required to fold the ribozymes and often for catalysis, usually prevents the dissociation of the newly synthesized strands from their template[3,9]. Thus, template-dependent RNA synthesis frequently leads to the formation of dead-end duplexes, which are incompatible not only with the release and folding of encoded ribozymes but also with the recycling of templates for further rounds of replication.

Temperature-induced melting appears as an attractive solution to separate product from template. However, high temperatures lead to substantial RNA degradation, especially at high magnesium concentrations[10,11]. These factors limit thermal denaturation to short strands in specific buffers and short heating times[12], possibly implemented by a centralized convection flow[13]. Strategies including toehold strand displacement[14] or nucleic acids with heterogeneous backbone chemistries[15,16] have been studied as potential solutions to this dilemma. Recently, RNA synthesis from a circular genome has been discussed as a solution to the strand separation problem[17] and limited synthesis of a catalytic RNA micromotif using rolling circle replication has been demonstrated[18]. Nonetheless, demonstrations of key reaction steps, particularly the general synthesis and release of more complex functional RNAs, are currently missing[11,19].

We previously explored geochemically plausible non-equilibrium systems that could overcome the template-inhibition challenge in the

[1]Systems Biophysics and Center for NanoScience (CeNS), Ludwig Maximilian University Munich, Geschwister-Scholl-Platz 1, 80539 Munich, Germany. [2]Department of Chemistry and Chemical Biology, TU Dortmund University, Otto-Hahn-Str. 4a, 44227 Dortmund, Germany. ✉e-mail: dieter.braun@lmu.de; hannes.mutschler@tu-dortmund.de

absence of excessively high temperatures[20]. Of particular interest are porous rock systems comprising heated air-water interfaces (hereafter referred to as AWI-systems), which can be experimentally reproduced in a defined manner by microfabrication. In this work, we demonstrate how AWI-systems allow ribozyme-catalyzed RNA replication of sense and antisense strands followed by subsequent strand-dissociation in a one-pot system. The combination of these reaction steps, which are otherwise mutually exclusive under isothermal conditions, enables the combined synthesis, release, and folding of active ribozymes. Overall, these results infer that abundant geothermal microenvironments had the potential to support replication and thus evolution of early biological systems.

## Results

### *sunY*-catalyzed RNA ligation at heated air-water interfaces

In $CO_2$-rich AWI-systems, nucleic acids are exposed to periodic changes in $Mg^{2+}$ concentration and pH level, the latter of which originates

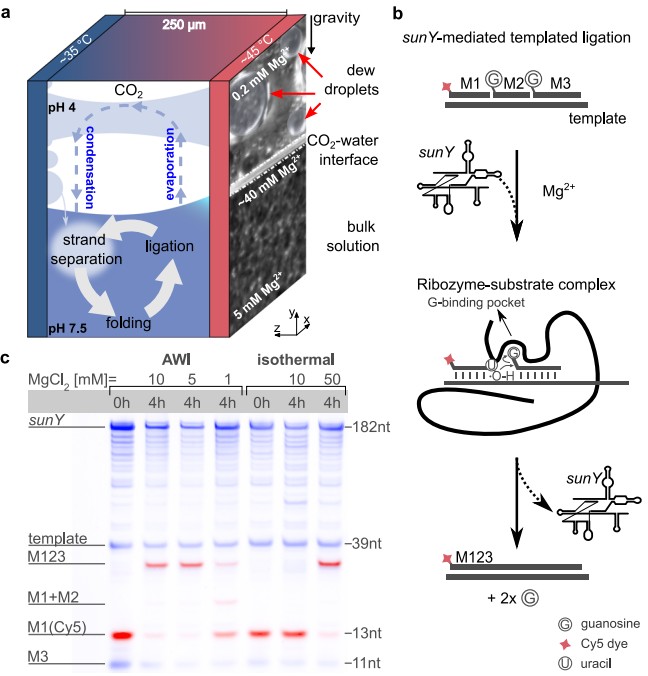

**Fig. 1 | Heated AWI-systems create micro water cycles that enable RNA folding, ligation, and strand separation at low magnesium concentrations.**
**a** Asymmetric heating triggers a temperature gradient across AWI compartments. This setup mimics a gas bubble inside a porous volcanic rock. RNA and ions are concentrated at the warm side of the interface by the evaporation of water. The evaporated water condenses as dew droplets on the cold side of the chamber. The droplets grow through surface tension and form a connection to the warm surface of the chamber, creating a miniature water cycle with strong fluctuations in salt concentration and pH similar to what has been described previously[20,21]. **b** The *sunY* ribozyme catalyzes templated ligation of RNA strands using guanosine nucleosides (G) as leaving group as schematically shown for the ligation substrates M1-Cy5 (red label), M2 and M3. The mechanism of ligation is based on the exon ligation step during RNA splicing by group I intron: after formation of the ribozyme substrate complex, the ligation junction of the substrate duplex is positioned in the catalytic center of the ribozyme by binding of the terminal guanine nucleotide of the downstream exon in the G-binding pocket[32]. The aligned 3'-OH group of the upstream exon performs an inline attack on the phosphodiester bond of the bound 5'-guanosine, leading to ligation of the adjacent upstream and downstream exons and release of the guanosine and the catalytic intron. **c** PAGE analysis (two channels: SYBR Gold, blue, and Cy5, red) shows that the AWI environment enables ligation of an RNA product M123 from three RNA substrates (M1-3) at $MgCl_2$ concentrations as low as 1 mM. To produce yields similar to 5 mM $MgCl_2$ in the AWI-system, a $MgCl_2$ concentration of 50 mM is required under isothermal conditions. Source data are provided as a Source Data file.

from an equilibrium of dissolved carbonic acid, bicarbonate, and carbonate in dew droplets depending on the applied partial pressure of $CO_2$[20,21] (Fig. 1a, Supplementary Fig. 1). While the low pH enables transient melting of otherwise stable nucleic acid duplexes presumably due to nucleobase protonation[22], the co-accumulation of RNA and magnesium ions at water-gas interfaces promotes folding and catalysis[23,24]. This accumulation is the result of the capillary flow created by water evaporation at the warm side of the air-water interface. The effect of water convection and Marangoni flow at the interface is minimal, but contribute if molecular assemblies grow to the size of tens of micrometers[23]. We, therefore, speculated about the ability of AWI-systems to provide a suitable environment for repeated RNA-dependent RNA replication. Derivatives of the self-splicing *sunY* intron from bacteriophage T4 can catalyze template-dependent oligonucleotide ligation using 5'-guanosines as a leaving group[4,5] (Supplementary Fig. 2a, b). Due to their robust activity and independence from prerequisite activation chemistries such as phosphate- or imidazole-based leaving groups, *sunY*-derived ribozymes are an attractive model system for primitive enzymatic RNA-replication[25,26]. Initially, we explored the $Mg^{2+}$ concentration requirements for RNA ligation by a 182 nucleotide (nt) variant of the *sunY* ribozyme (Supplementary Fig. 2c) in $CO_2$-rich AWI-systems (Supplementary Figs. 1, 3, 4). To this end, we probed the template-dependent ligation of a 30 nt RNA from three oligonucleotide substrates (Fig. 1b, Supplementary Table 1, Supplementary Note 1): a 5'-Cy5 labeled substrate (M1, 13 nt), which allows direct fluorescence-based PAGE analysis, and two downstream 5'-guanosine 'activated' substrates, M2 (7 nt), and M3 (10 nt). Due to the concentration effect at the evaporation zone of the heated side of the air-water interface, a significantly lower $Mg^{2+}$ bulk concentration (5 mM) was required to observe near-complete substrate ligation ratios (89%) compared to isothermal conditions (50 mM, 86%) after 4 h of incubation (Fig. 1c). We observed ligation in the AWI-system even at bulk concentrations as low as 1 mM $MgCl_2$ (17%), whereas no detectable ligation occurred in a comparable equilibrated system at constant temperature (Fig. 1c, Supplementary Fig. 5). These findings confirmed that the local up-concentration of solutes at the warm side of the AWI interface was sufficient to allow *sunY*-dependent RNA ligation at bulk magnesium concentrations considerably lower than those required under isothermal conditions.

### AWI-systems enable *sunY*-catalyzed ligation of *sunY* fragments

We hypothesized that the periodic local pH decreases, resulting from the precipitation of acidified dew droplets back into the bulk reservoir (Supplementary Fig. 6), could lead to a transient decrease in RNA melting temperatures thereby promoting the release of ligation products from their template and allowing intramolecular folding into functional RNAs. This encouraged us to attempt to synthesize strands with the same or similar sequence as *sunY*. The Szostak group previously demonstrated that an active version of *sunY* can assemble non-covalently from three oligonucleotides (A, B, and C) between 43 and 75 nt in length[4]. We reasoned that the AWI-system might circumvent template inhibition during templated RNA ligation and therefore explored if AWI-based non-equilibrium environments could also support replication of *sunY*-derived RNA strands. We initially explored if AWI-systems enable *sunY* to catalyze the synthesis of each of the three fragments A, B, and C from three short oligonucleotides substrates (A1-3, B1-3, and C1-3) (Supplementary Table 1, Supplementary Note 1). To this end, we probed the ligation of each fragment (herein referred to as A123, B123, and C123) from three substrate strands starting from equal concentrations (2.5 μM) of *sunY* and template and fourfold excess of each ligation fragment (Fig. 2a). We observed good yields of full-length products (38.5 ± 1.5% A123, 19.5 ± 2.5% B123 and 22 ± 7% C123) after 2 h of reaction in the AWI-system and only minor amounts of incomplete intermediates. In contrast, we observed higher relative amounts of intermediate products and only low full-length yields for the three

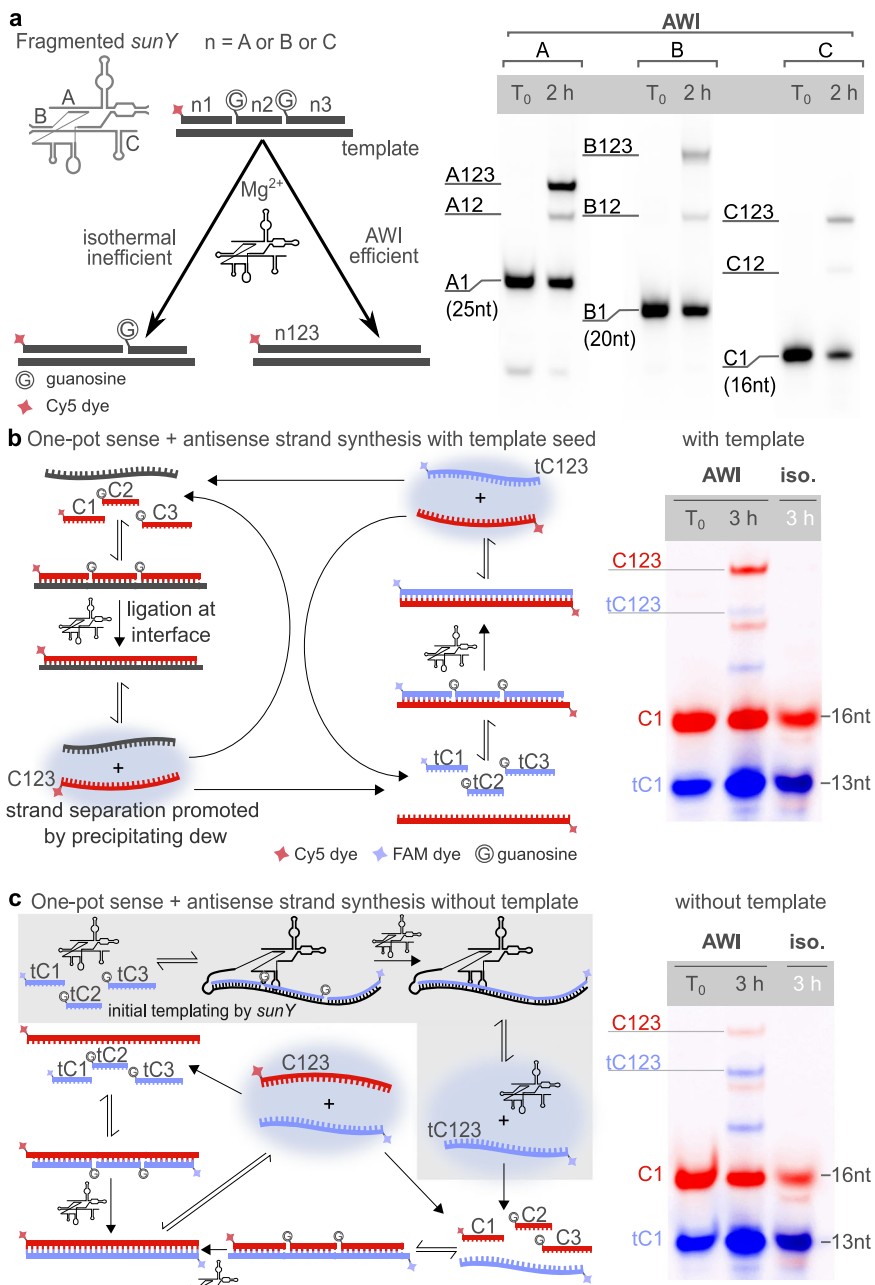

**Fig. 2 | Complete replication cycles of sunY fragments. a** Illustration (left) and PAGE analysis (right) of separately performed template-dependent ligations of the three *sunY* fragments A123, B123, and C123. Lanes show reactions before (T$_0$) and after 2 h of incubation. **b–c** Full replication cycle of fragment C123. Replication proceeds vial ligation of the RNA substrates for the sense strand (C1, C2, C3-Cy5, red) and the antisense strand (tC1-FAM, tC2, tC3, blue). In the AWI-systems, both the sense and antisense strands emerged **b** with the template present and **c** without the template present, suggesting that *sunY* partially acted as a template for antisense strand synthesis (gray box). Under isothermal conditions no products were observed after PAGE analysis. Source data are provided as a Source Data file.

fragments under isothermal conditions (~8% A123, ~1% B123, and ~1.3% C123 after 2 h), indicating that the high melting temperatures of the different RNA duplexes in the system might limit processivity under bulk conditions (Fig. 2a, Supplementary Fig. 7, Supplementary Table 2).

**Coupled sense and antisense synthesis of *sunY* fragments**

After demonstrating that all three *sunY* fragments can be synthesized by the full-length ribozyme in situ, we sought to explore if AWI-systems can also support full cycles of RNA replication, synthesizing both sense and antisense strands of its fragments in a single reaction environment. To this end, we carried out separate ligation experiments using

substrate oligonucleotides for the *sunY* fragments C123 and A123 as well as their corresponding templates, e.g., tA1, tA2, tA3 for tA123 (Supplementary Table 1, Supplementary Note 2). To initiate ligation of the sense fragments, we included seed amounts of tA123 and tC123 in each reaction. We detected the formation of both sense (C123, A123) and antisense strands (tC123, tA123) after 3 h of incubation suggesting that once formed, both A123 and C123 serve as templates for the synthesis of their respective antisense templates tA123 and tC123. (Fig. 2b, Supplementary Fig. 8a, b). Intriguingly, we also detected the formation of both sense and antisense fragments even in the absence of seeding template (Fig. 2c, Supplementary Fig. 8), suggesting that the *sunY* ribozyme itself can act as template for the synthesis of A123 and

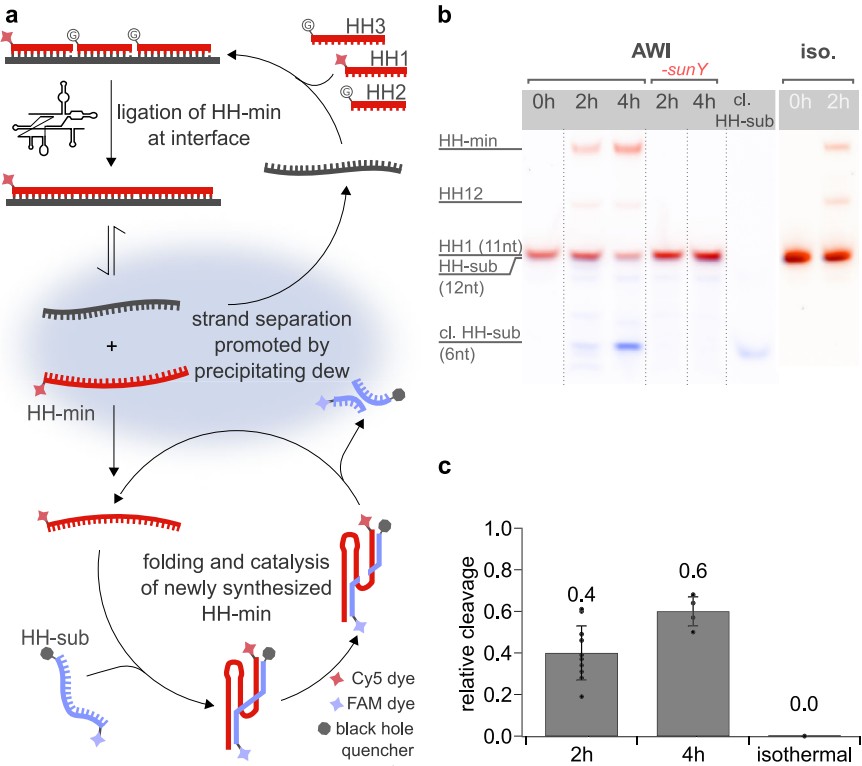

**Fig. 3 | Templated synthesis and release of active hammerhead ribozyme.**
**a** *sunY* catalyzes template-dependent ligation of the hammerhead ribozyme (HH-min, red) from the three RNA substrates (HH1-3, red). Local variations in pH and salt concentration cause strand separation, releasing ligated HH-min, which can then bind to and cleave its substrate (blue). **b** Ligated HH-min (red, Cy5) was observed in both the non-equilibrium and isothermal environments after 2 h of incubation. Formation of cleaved hammerhead substrate (HH-sub, blue, FAM) was observed only after incubation under non-equilibrium (AWI) conditions. Control experiments without *sunY* ribozyme exhibited no HH-min assembly or HH-sub cleavage,

demonstrating that the combined unligated HH-min fragments HH1-3 have no residual RNA-cleavage activity. The isothermal control also showed no activity. Individual gel lanes (outlined with dotted lines) have been rearranged for clarity. **c** Bar graph showing the calculated yields of HH-sub cleavage after 2 h and 4 h in the AWI-system and under isothermal control conditions. Data are presented as mean values ± SEM. Black circles represent the values from independent replicates with $n = 10$ for 2 h incubation and $n = 4$ for 4 h incubation. Source data are provided as a Source Data file.

C123, which subsequently can serve as template for the formation of tA123 and tC123.

## One-pot synthesis and folding of active hammerhead ribozyme

We speculated that release of the ligation products from their template might lead to folding of the products into functionally active RNAs, a process similar to transcription. To test this hypothesis, we designed a *sunY*-based assembly assay for a minimal version of the hammerhead ribozyme (HH-min) derived from the tobacco ring spot virus satellite RNA (Fig. 3a)[27,28]. In this system, HH-min is synthesized by *sunY* from three RNA substrates (HH1-3, 11 nt, 11 nt, 16 nt), with ligation junctions interrupting critical catalytic motifs of HH-min, such that catalysis of the fragmented HH-min was prevented (Supplementary Fig. 2d, Supplementary Table 1, Supplementary Note 1). While experiments in both the AWI-system and under standard isothermal conditions led to the synthesis of HH-min, the AWI-system promoted the synthesis of HH-min even at a 10-fold lower MgCl$_2$ concentration (5 mM) than required for isothermal ligation (50 mM, Supplementary Fig. 9), corroborating our previous data regarding magnesium dependence under non-equilibrium conditions. Kinetic measurements for both conditions further revealed that the AWI-system indeed enabled multi-turnover synthesis of HH-min as the resulting yields of the full-length HH-min, which peaked after about 4 h of incubation, exceeded the amount of input template RNA (Supplementary Figs. 10–12). In contrast, no excess synthesis of HH-min over template could be observed under isothermal conditions, indicating that all product strands remained tightly associated with the template. The

observation of multi-turnover ligation in the AWI-system suggested the presence of single-stranded, de novo synthesized HH-min in solution. To probe if this dissociated RNA species was catalytically active, we repeated the experiment in presence of the cognate HH-min substrate (HH-sub) with the aim of observing catalytic substrate cleavage. Satisfyingly, PAGE analysis of the reaction mixture confirmed that HH-min formation was accompanied by cleavage of HH-sub under non-equilibrium conditions (cleavage ratio of 40 ± 13% and 60 ± 7% after 2 h and 4 h, respectively), validating that product dissociation, folding, binding and cleavage occur in addition to *sunY*-catalyzed ribozyme assembly in a one-pot system (Fig. 3b, c, Supplementary Table 3). In contrast, no HH-sub cleavage was observed under isothermal conditions, suggesting that the newly synthesized HH-min was unable to dissociate from the template under these conditions (Supplementary Fig. 13). This interpretation agrees with the predicted melting temperature of the HH-min-template complex at the magnesium concentrations used in the reaction (90 °C, Supplementary Table 4). As expected, control experiments in the AWI-system without *sunY* resulted in no observable HH-min assembly or HH-sub cleavage.

## Discussion

This work revealed a credible model prebiotic reaction environment that is compatible with purely RNA-catalyzed replication of short functional RNAs. Similar non-equilibrium microenvironments might have been able to host the first autocatalytic nucleic acid replicators that marked the onset of biological evolution towards higher complexity. In its current form, AWI-systems are well suited to enable

combined replication and strand-dissociation of RNA strands between 20 and 75 nt from much shorter precursor oligonucleotides, which have the potential to assemble into more complex ribozymes. Typically, fragmented ribozyme variants are in most cases less active than their full-length counterparts presumably due to their lower stability and folding defects. In line with this, the fragmented *sunY* variant formed by noncovalent assembly of A123, B123, and C123, unlike the full-length variant, did not show detectable ligation of HH-min under standard AWI or isothermal conditions (Supplementary Fig. 14a).

Interestingly, however, we found that lowering the temperature on the warm side of the AWI-system from 45 °C to 40 °C allowed the fragmented *sunY* ribozyme to synthesize moderate amounts of HH-min, illustrating that milder conditions could compensate to some extent for the lower activity of the split variant (Supplementary Fig. 14a). To explore the autocatalytic potential of the system, we demonstrated the synthesis of all three *sunY* fragments (A123, B123, and C123) directly by the fragmented *sunY*, although the yields were considerably lower than for ligations catalyzed by the full-length ribozyme (Supplementary Fig. 14b–d). Further optimization will be required to improve reaction yields, but the results provide an optimistic outlook on the capability of the system to undergo self-replication from a pool of shorter oligonucleotide substrates.

AWI-systems can be adapted to the requirements of other ribozyme systems by altering temperatures or chamber geometries. Of particular interest are RNA polymerase ribozymes that catalyze templated primer extension reactions of equal sequence length to themselves. As previously mentioned, the problem of separating de novo RNA from its template is exacerbated by salts in solution. This is even more the case for RNA polymerase ribozymes that have been characterized under magnesium concentrations ranging from 50–200 mM[6–8]. It remains to be seen how these systems also benefit from the salt and pH oscillations to enhance the yields by the positive feedback of template release and by reduced metal ion- and pH-mediated degradation of catalytic RNAs.

In conclusion, the data presented suggest that the heated air-water interface (AWI) system has the potential to host fragmented RNA replicators that catalyze both general and autocatalytic template-directed RNA synthesis, further strengthening the hypothesis that dynamic, non-equilibrium environments were cradles for the emergence of higher order biomolecular complexity. Intriguingly, similar interface settings can drive other prebiotically relevant key processes such as the phosphorylation of mononucleosides, polymerization of RNA, encapsulation of nucleic acids in membrane vesicles as well as the fusion and fission of coacervate-based protocells[20,23,29,30]. Future work might explore the suitability of AWI-systems to combine these key aspects of biological systems into a single collective scenario providing a setting that could potentially support most, if not all, of the required capabilities for an inanimate system to adopt life-like behaviors.

## Methods
### Nucleic acids
RNA oligonucleotides were ordered from Biomers or Integrated DNA Technologies (IDT) in dry form and subsequently adjusted to 200 μM stock concentrations with nuclease-free water (Ambion™ Nuclease-free water from Invitrogen). HH-sub was synthesized by eurofins Genomics. Final RNA concentrations were confirmed using 260 nm absorbance. All stocks were kept at −80 °C and thawed on ice prior to the experiment. Filling and extraction of AWI-systems were performed on ice. The *sunY* DNA template for in vitro transcription was ordered as a gBlock with the following sequence:

GATCGATCTCGCCCGCGAAATTAATACGACTCACTATAGGGAAA ATCTGCCTAAACGGGGAAACACTCACTGAGTCAATCCCGTGCTAAAT CAGCAGTAGCTGTAAATGCCTAACGACTATCCCTGATGAATGTAAGG-GAGTAGGGTCAAGCGACCCGAAACGGCAGACAACTCTAAGAGTTGAA GATATAGTCTGAACTGCATGGTGACATGCAGGATC

The gBlock was amplified using the following primers:
$P_{rev}$ = GATCCTGCATGTCACCATGCAGTTCAGACT;
$P_{fwd}$ = GATCGATCTCGCCCGCGAAATTAATACGAC

All RNA oligonucleotide sequences are provided in Supplementary Table 1.

### Transcription of *sunY*
The *sunY* DNA template was amplified by PCR using the Q5 High Fidelity 2x Master Mix (NEB) with 10–25 ng gBlock template and 1 μM $P_{fwd}$ and $P_{rev}$. The PCR protocol was: 30 s at 98 °C, 12 cycles of 7 s at 98 °C and 50 s at 72 °C, followed by a 2 min step at 72 °C. The reaction was cleaned using the Monarch PCR cleanup oligo kit (NEB) and eluted in 6-10 μL of nuclease-free water. The transcription buffer consisted of 40 mM Tris·HCl pH 8, 20 mM MgCl$_2$, 10 mM DTT and 2 mM spermidine. Template (approximate final concentration of 0.5 μM) and 3.75 mM of each NTP, 5 U/mL or 2 U/400 μL of *E. coli* Inorganic Pyrophosphatase (100 U/mL stock) and 1500 U of T7 polymerase were added. The transcription was incubated for 3–4 h at 37 °C, and subsequently cleaned using the Monarch RNA cleanup kit (NEB). The resulting material was gel purified using a 2 mm 15% PAGE gel (30 mL) with 10% stacking gel (10 mL), (6 well comb, 25 W constant power for 2 h). The gel was wrapped in plastic wrap, placed on TLC plate, and illuminated under UV (254 nm). The ribozyme band was marked, excised, placed in a 2 mL tube and weighed. The gel slice was dry crushed with a syringe plunger, 0.3 M sodium acetate pH 4.8–5.2 was added (2× mg gel weight in μL) and soaked overnight at 4 °C on a rotator. The next day, the supernatant was recovered using Spin-X centrifuge tube filters (Corning, 0.45 μm, sterile) and centrifuging for 5 min at maximum speed. Subsequently, 20 ng glycogen and either 1 volume isopropanol or 2.5 volumes of ethanol were added, and the sample was frozen for 45 min to 1 h (−80 °C for ethanol, −20 °C for isopropanol) and centrifuged for 45–90 min at −9 °C, maximum speed. The supernatant was removed, and the pellet washed with 80% ethanol and centrifuged again for 5 mins at maximum speed. This step was repeated twice before drying the pellet for 10–15 min. Finally, it was dissolved in deionized H$_2$O and concentrations were determined by measuring the absorption at 260 nm using a Nanodrop (Thermo). Extinction coefficients for sequences were calculated using OligoCalc[31].

### Ligation of M123
RNA ligation reactions were carried out with final concentrations of 20 μM for all ligation substrates, 5 μM of template, 30 mM Tris·HCl pH 7.5, 100 mM KCl, and varying bulk magnesium concentrations (as specified). Ligation was initiated by adding *sunY* ribozyme to a final concentration of 2.5 μM. The first substrate (M1) contained a Cy5 label at the 5′-end. A reaction volume of 12 μL was prepared for the isothermal control and 17 μL were prepared for the experiments under AWI conditions. As soon as *sunY* was added, 2 μL of each sample was quenched with 2× gel loading buffer (90% formamide, 5% glycerol, 50 mM EDTA, 0.015% Orange G). For specifics regarding the PAGE protocol, please refer to subsection 'Polyacrylamide gel electrophoresis (PAGE)'. The remaining volume for the isothermal controls was placed in the T100 thermal cycler (Bio-Rad) set to 45 °C. The reaction volume of 15 μL for the AWI experiments was injected into the fully assembled AWI-system through a Teflon tubing (Techlab) using a Hamilton syringe. The reaction time was 4 h. Please refer to the subsection 'Preparation of AWI chambers and sample injection' for details of the AWI assembly and filling process.

### Synthesis of *sunY* fragments A123, B123, and C123
The synthesis reactions took place for 2 h at a final concentration of 10 μM of each ligation substrate (including the 5′ Cy5 labeled 5′-substrate) and 2.5 μM of template, respectively. The reaction buffer was 30 mM Tris·HCl pH 7.5, 100 mM KCl, and 10 mM MgCl$_2$ and all

reactions were initialized by the addition of *sunY* to a final concentration of 2.5 μM. Setup of reactions under AWI/isothermal conditions was performed as described above.

## Complete replication of *sunY* sense and antisense fragments

For a full replication cycle of a *sunY* fragment, ligation substrates for both the sense and antisense were added. Syntheses of A123 and tA123 were carried out with final concentrations of 10 μM for each substrate (A1, A2, A3) and the complementary substrates (tA1, tA2, tA3). Reactions were incubated for 3 h with and without 2.5 μM of template. The first substrate of the A123 and C123 ligation reactions (A1 / C1) were 5′Cy5 labeled, and the 5′-terminal substrates of the tA123 and tC123 ligation reactions (tA1 / tC1) were 5′FAM labeled. For the synthesis of C123 and tC123 the concentrations had to be adjusted slightly to decrease the overall concentration of strands complementary to the full-length *sunY* ribozyme as this increased yields and reduced incubation times. Here, four concentration ratios were tested and are denoted in the following by "Template: complementary substrates: substrates (T:tS:S)": 2.5 μM: 5 μM: 10 μM (Fig. 2b, c; Supplementary Fig. 8c); 1.5 μM: 2.5 μM: 10 μM; 1.5 μM: 5 μM: 10 μM and 2.5 μM: 2.5 μM: 10 μM (Supplementary Fig. 8b). All replication reactions were performed in 30 mM Tris·HCl pH 7.5, 100 mM KCl and 10 mM MgCl₂. Substrate ligation was initialized by the addition of *sunY* to a final concentration of 2.5 μM. For the ligation of C123 and tC123, the experiments ran for 3 h or 18 h as indicated.

## HH-min synthesis

HH-min synthesis reactions took place at a final concentration of either 10 μM and 5 μM or 12 μM and 3 μM of all ligation substrates and template, respectively. The first RNA ligation substrate (HH1) was either 5′FAM or 5′Cy5 labeled. The buffer in the AWI reactions and isothermal controls was 30 mM Tris·HCl pH 7.5, 100 mM KCl with varying bulk magnesium concentrations ($c_{bulk}$—as specified). Reactions were initialized by the addition of *sunY* to a final concentration of 2.5 μM. To verify the activity of the in situ synthesized HH-min hammerhead ribozyme, 0.4 μM HH-sub was added. The HH-sub contained 5′-terminal FAM tag and a 3′ BHQ quencher. For a $T_0$ sample, 2 μL of the prepared reaction was quetched immediately with 18 μL gel loading buffer (94% formamide, 5% glycerol, 0.01% Orange G) after initialization of the reaction. All isothermal controls were incubated at 45 °C. To verify that the three unligated HH-min ligation substrates HH1-3 had no residual RNA-cleavage activity that was e.g. triggered by the accumulation process at the water-air interface, 2 h and 4 h control reactions in the absence *sunY* were performed (Supplementary Fig. 9).

## Polyacrylamide gel electrophoresis (PAGE)

All samples were analyzed using polyacrylamide gel electrophoresis (PAGE) with 15% acrylamide. The gel stock was prepared from the Roth Rotiphorese DNA sequencing gel stocks (acrylamide:bisacrylamide = 19:1), where a 0.75 mm thick gel with a 15-tooth comb needed about 5 mL gel mixture, containing 3 mL gel concentrate, 1.5 mL gel diluent, 0.5 mL buffer concentrate, 25 μL APS and 2.5 μL TEMED. All gels were run in denaturing conditions of 50% urea and 1× TBE buffer at -50–55 °C. Each runtime consisted of a 30 min pre-run at 400 V, after which 4-5 μL were loaded per well. 1:9 dilutions of samples with gel loading buffer (94% formamide, 5% glycerol, 0.01% Orange G) were used for PAGE analyses of $T_0$ samples and samples from isothermal experiments and 3:7 dilutions were used for PAGE analyses of samples from AWI experiments. Electrophoresis was performed for 5 min at 50 V followed by 18–24 min at 300 V. After each run, gels were either directly imaged or stained in 50 mL of 1× TBE buffer with 5 μl of 10,000× SYBR Gold Nucleic Acid Gel Stain (Thermo Fisher Scientific) for 5 min prior to imaging. Imaging was done with a two-channel protocol of Image Lab v6.0.1 from the Bio-Rad ChemiDoc MP System, using the blots for Cy5 and Alexa488. All gel images were imported

into an in-house written LabView routine (NI LabVIEW 2014 14.0.1f11 (64 bit)) for densitometric analysis. Recorded lane intensities were background-corrected using the average intensity of the inter-lane spaces. Each band was fitted with a Gaussian in Igor Pro 6.37. As the extraction from AWIs as well as quenching from imaging during the experiment can introduce artifacts, only relative intensities for each lane were used for quantification:

$$r = \frac{I_{Product}}{I_C + I_{intermideate} + I_{Bg} + I_{Product}} \quad (1)$$

Intensities from the uncleaved substrate HH-sub were multiplied by 1.69, which is the factor by which the in-gel fluorescence is quenched due to the presence of the Black Hole Quencher at the 3′end.

## Preparation of AWI chambers and sample injection

AWI chambers were built similarly to the trap system reported in Matreux et al.[24]. A schematic image of a chamber is shown in Supplementary Fig. 1. The desired shape was cut out from a 250 μm thick Teflon foil (HolscotEurope) with using a GRAPHTEC CE6000-40 Plus cutter. The resulting Teflon slide defined the shape of the non-equilibrium AWI compartments. The Teflon cut-out was designed to have three reaction compartments. The compartments are all 11 mm wide and 8 mm high. This corresponds to a maximum capacity of 22 μL for a 250 μm foil thickness. A second foil with 100 μm thickness was cut out to serve as hydrophobic backside. Both Teflon slides were sandwiched between two sapphire windows and fixed to a metal block using a steel frame. To increase heat conductivity between the cold sapphire and the metal surface, a 25 μm thick graphite (PANASONIC) foil was sandwiched in between both elements. All parts of the sandwich had a width of 22 mm and a height of 60 mm. Prior to assembly, all components were cleaned with ethanol and milliQ water and additionally cleaned from dust using scotch tape. A heater was screwed onto the top side. To achieve the best heat conductivity between the heater element and the surface of the AWI chamber, 200 μm thick graphite foils (PANASONIC) were placed between heater and the assembled AWI-system, as well as between the AWI-system and the water-cooled backside of the setup. The temperature gradient was maintained by cooling the backside with a refrigerated water bath (CF41 and 300 F, Julabo) and heating the front with an electrical heater regulated by an Arduino microcontroller board. For each experiment, a reaction volume of 15 μL was injected with a Hamilton syringe through a Teflon tubing (Techlab), which were fixed by PTFE fittings (Techlab). The inlets were then sealed with soft silicon and PTFE fittings. All compartments were connected to the same CO₂ inflow to ensure similar pressure conditions. Details of the resulting dynamics are described in Supplementary Note 3. Additionally, to monitor the temperature by the water bath and set the temperature of the heater, a temperature sensor was screwed into one of the holes in the backside with one of the fittings.

To create the non-equilibrium conditions in the AWI-system, two types of setups were used either with or without the possibility of optical readout. The air phase of each AWI-system was connected to a CO₂ supply and the pressure was adjusted by a dedicated pressure controller (RIEGLER) directly at the setup. Additionally, a manometer (Bourdon Instruments, MEX 3) was attached to the CO₂ supply to cross-verify the applied pressure. Details of the influence of the CO₂ atmosphere on the pH of the bulk reaction are described in Supplementary Methods 3.1. Temperatures were monitored throughout the experiment using a GTH 1170 thermometer (Greisinger) at the back sapphire. Temperatures were further monitored at the beginning and end of each experiment with a heat camera (Seek Thermal, SQ-AAA) on the front sapphire to monitor the applied gradient. The measured temperatures were used as input parameters for a finite-element simulation to simulate the thermal gradient inside the chamber.

Details of the temperature simulation are provided in Supplementary Fig. 4 and Supplementary Methods 3.2. For fluorescence excitation, the THORLAB LEDs M470L2-C4 (470 nm emission) and M625L2 (625 nm emission) were used for the dyes FAM and Cy5, respectively. Imaging was performed with the Axio Scope.A1 microscope (Zeiss) with infinity corrected long working distance objective (2× lens Mitotoyo, M Plan Apo) and a 0.5 adapter that was coupled to a stingray camera F145B ASG (Allied). Temperature and microscope control were realized with code written in NI LabVIEW 2014 14.0.1f11 (64 bit).

## Statistics and reproducibility

The gel presented in Fig. 1c is a representative from $n = 4$, $n = 4$, and $n = 2$ independent replicates in the AWI-system for MgCl$_2$ concentrations of 10 mM, 5 mM, and 1 mM, respectively, and $n = 2$ and $n = 5$ for MgCl$_2$ concentrations of 10 mM and 50 mM in isothermal conditions. All corresponding gels are provided in the Source Data file. In Fig. 2a the shown gel is representative for $n = 2$ (A, B fragment) and $n = 3$ (C fragment) independent experiments for reactions in the AWI-system. The remaining gels are shown in Supplementary Fig. 7 and are provided in the Source Data file. The gels shown in Fig. 2b, c represent a single concentration ratio ($n = 2$ independent experiments), which was selected based on several pre-tested concentration ratios (each $n = 1$). The remaining concentration ratios are shown in Supplementary Fig. 8. All gels are provided in the Source Data file. In Fig. 3b the gel lanes were reordered for clarity as indicated by the dotted line. The unmodified gel is shown in Supplementary Fig. 13B. The isothermal control for this experiment was run on a different gel. The 2 h and 4 h lane (+*sunY*) of the AWI-system are representative gels from $n = 10$ and $n = 4$ independent experiments, respectively. These replicates are the source data for Fig. 3c and are provided in the Source Data file. Details on the statistics and reproducibility of the data shown in the Supplementary figures are provided in Supplementary Methods 3.3.

## Reporting summary

Further information on research design is available in the Nature Portfolio Reporting Summary linked to this article.

## Data availability

The data generated in this study are provided in the paper and Supplementary Information file. Figs. 1–3 and Supplementary Figs. 3–5, 7–14 contain associated raw data provided in the Source Data file and deposited at https://doi.org/10.5282/ubm/data.362. Source data are provided with this paper.

## Code availability

Codes are provided at https://doi.org/10.5282/ubm/data.362.

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

## Acknowledgements

The authors thank Christof B. Mast for help with the setup and for providing his expertize on the non-equilibrium compartments and Alan Ianeselli for help with the $CO_2$ installation. We also thank Borislav Mladenov for their experimental support. Financial support was provided by the European Research Council (ERC Evotrap, grant no. 787356, D.B. and RiboLife, grant agreement no. 802000, H.M.), the Simons Foundation (grant no. 327125, D.B.), the Deutsche Forschungsgemeinschaft (DFG, German Research Foundation) grant TRR 235 (CRC 235) (Project-ID 364653263, A.S., D.B) the Excellence Cluster ORIGINS (D.B.) funded by Germany's Excellence Strategy EXC-2094-390783311, and the Center for NanoScience (A.S., D.B.). H.M., D.B., and K.L.V. acknowledge support from the Volkswagen Foundation. AS is supported by the Add-on Fellowship of the Joachim Herz Foundation.

## Author contributions

Project design, funding acquisition, and supervision: D.B., H.M.; experiments and data analysis: A.S., L.K., E.S., K.L.V.; visualization: A.S., E.S.: writing: A.S., L.K., E.S., K.L.V., D.B., H.M.

## Funding

## Competing interests

The authors declare no competing interests.
