## [Peer Review File · Nature Communications]

Ribozyme-mediated RNA synthesis and replication in a model
Hadean microenvironmentREVIEWER COMMENTS

Reviewer #1 (Remarks to the Author):

I think it's a well written paper, although it is hard to discern the major novelty points of this work.

The experimental plan and rationale is very clear, has a place and the system is not needlessly complicated. The paper is well written, the narrative flows well and all data is presented in a convincing way. The figures are clear. The experiments are concise and streamlined.

I am unclear if the Authors actually make a dew-drop forming system?

The descriptions suggests Authors just change the temperature and Mg concentration? Most of the time, the paper says "AWI conditions" or "AWI-like system," but sometimes it is "AWI system," which doesn't seem accurate unless they're remaking the rock-water interface from Fig1A.

It would be useful to clarify that and make it consistent through the text.

How artificial having the G-leaving groups are? Are they truly prebiotically plausible?

If not, still a really interesting and elegant paper, but it requires clarification.

Either way, I think it would be very beneficial to the readers, placing it in the broader context of prebiotic evolution, to address how plausible their starting structures are pre-biotically. Perhaps also it would be useful to discuss what needs to be furthered to either make these G-activated RNAs or if there's any systems that don't need them.

While this is well written paper with elegant experiments, I am not sure if this is an advancement significant enough for a major publication.

All elements of the system described here were presented before, and this work doesn't add a significant advancement to our understanding of prebiotic replication, hydrothermal cycling or ribozyme biochemistry.

Reviewer #2 (Remarks to the Author):

Summary

The manuscript Salditt et al. explores how the environment of a temperature gradient microchamber - as model system for a Hadean rock pore - can enhance, and facilitate processes related to the replication of catalytic RNAs. It combines three phenomena that were published previously (the temperature gradient induced formation of pH gradients that lead to destabilization of RNA duplexes, the temperature gradient induced localization of Mg²⁺ / RNA to the gas / water interface that lead to ribozyme activity under otherwise suboptimal Mg²⁺ concentrations, and the templated ligation of a hammerhead ribozyme by a group I intron (I'm not sure whether the latter is published - but it would not be a significant advance in itself)), and the novelty of this manuscript lies in the combined effect of these three phenomena. While there is likely also formation of a complete sunY ribozyme, this formation is not confirmed, which could be done either by reverse transcription and sequencing of several clones, or by showing catalytic activity of the sunY ribozyme.

The concentration effect of RNA and Mg²⁺ at the gas/water interface allowed template-directed, ribozyme mediated ligation of three short oligonucleotides to occur at bulk Mg²⁺ concentrations that otherwise do not support efficient catalysis. Additionally, the effect of the CO₂ atmosphere generated a local pH decrease from dissolving CO₂ in dew droplets of the temperature gradient microchamber, which allowed strand separation of the products in detectable quantity. When the hammerhead ribozyme was formed by group I intron mediated, templated ligation, catalytically active hammerhead ribozyme was produced.

I feel that the study may be publishable in Nature Communications if either replication is demonstrated as described below, or if the title, abstract, and text are toned down to correctly describe the findings.

Major points

1 - The title and abstract is misleading, in two ways; the first is most important.

First, there are no RNA replication cycles in this study, especially no complete ones. The manuscript shows that a sun Y ribozyme mediates the templated synthesis of a functional hammerhead ribozyme from three fragments. This is not replication, this is ligation / synthesis. For replication, at a minimum, both strands of the hammerhead ribozyme would have to be produced, and it would have to be shown that the two strands template each other's synthesis. Alternatively, the synthesis of both strands of the subY ribozyme by the sunY ribozyme could be shown. So far, this is not contained in the manuscript because only one strand seems to be produced, and because the sequence of the ligation product is not confirmed. The latter could be done either by sequencing of the ligation products (at least 3 clones with correct sequence) or demonstrating catalytic activity of the product strand. The incorrect impression of replication in the title is enhanced by the word 'complete' in the title.

Second, while the temperature gradient chamber is a model system for a Hadean microenvironment it cannot be used to say that it *is* a Hadean microcompartment (in the absence of direct Hadean evidence). If there is no direct evidence that this microcompartment (with such a temperature gradient) existed in the Hadean then it needs to tone down such as 'Hadean rock crack model chamber' or get more specific such as '... in a CO₂-rich atmosphere and a temperature gradient microchamber as model system for Hadean rock pore'.

3 - Line 59 and 60-62: Since the presented system is based on several phenomena that are now well known in the wider scientific community, their background needs to be explained in more detail:

(i) The equilibrium that leads to the local low pH needs to be explained with a bit more detail, and more accurately. 'Carbonate' by definition is the (CO₃)²⁻ divalent anion - but this almost certainly does not exist under the conditions described. In contrast, this seems to be an equilibrium between dissolved CO₂ / carbonic acid and hydrogen carbonate, where the equilibrium is around pH 6.4.

(ii) the ability of pH around 4 to partially melt duplexes, and

(iii) the accumulation of RNA and Mg²⁺ at the water/gas interface need to be explained, each with perhaps one sentence.

[(iv) the sun Y ribozyme and its catalyzed reaction is explained satisfactorily]

Minor points

Line 15: 'stored on' -> 'stored in'

Line 16-17: There is at least one additional hurdle why 'experimental systems for sustained RNA-dependent RNA-replication are difficult to realize': Only very few systems are available, and all of them are quite limited (I know of four: van Kiedrowsky's replicating hexanucleotide, Joyce's replicating R3C ligase ribozyme, Joyce's cross-replicator, and replicating group I introns, one of which that is the model in this study). I suggest either (i) inserting a 'partially' / 'in part' such that it reads '... are difficult to realize, in part due to the high ...' or (ii) include additional hurdles such as the sparsity of designed and/or selected RNA systems described above.

Line 20-21: The abstract should include the information where these salt and pH oscillations come from (since there was no Hadean experimenter who added salt / water and acid / base). For example, "wet / dry cycling that resulted in oscillations of salt concentrations ...".

Line 33: Not all ribozymes require high magnesium concentrations. For example, group I introns are fine with 5 mM Mg²⁺, and some self-cleaving ribozymes can even work completely without Mg²⁺. Please tone down. Alternatively, if specific ribozymes are envisioned (see my comment to lines 16/17), please list them so that the reader knows what systems are discussed (without looking at the references).

Line 77 and 81'.. concentrations..." -> " ... bulk concentrations ..." because the concentration at the gas/water interface is significantly higher, which allows catalysis to occur.

Line 80: The word 'drive' is incorrect here because the driving force of the reaction is the leaving of the 5'-guanosine leaving groups. I suggest using 'mediate' or 'facilitate'.

Fig. 3: The strand of HH-min and the three fragments HH1-3 should be explicitly labeled in the figure since HH-min and HH1-3 have the same color. It becomes clear after making sense of the figure - but it should be clear without first understanding the figure.

Reviewer #3 (Remarks to the Author):

The manuscript by Salditt et al. reports an experimental scenario of sustained RNA replication at heated air-water interfaces in a CO₂-rich atmosphere. It is a very nice work that contributes to the solution of a long-standing problem in the study of the origin of life, i.e. product inhibition during copying of a nucleic acid strand due to the high thermodynamic stability of the duplex formed.

Using a sophisticated experimental setup, the authors show that oscillations in salt concentration and pH drive the destabilization of the duplex, allowing iterative cycles of copying and replication of functional RNAs. The paper is well written, the data presented are reliable, and the conclusions drawn are reasonable and supported by the work.

I have only a few minor comments that should be considered before publication:

- 1) SunY was used as an RNA ligase. Although sunY is shown schematically and also in sequence (Fig. S2), it would be helpful to show how the template-substrate complex interacts with the ribozyme to become ligated. Since not all readers are familiar with the exact mechanism of sunY, it might also be useful to elaborate on the catalytic center with the ligation site and the position of the guanosine residue that acts as the leaving group in this process.

2) On page 4, beginning at line 82, it states, " We hypothesized that the periodic local pH decreases, resulting from precipitation of acidified dew droplets (Figure S6), could lead to a transient decrease in RNA melting temperatures thereby promoting the release of ligation products from their template and allowing intramolecular folding into functional RNAs." Has the exact pH at the air-water interface where replication occurs been determined? The melting temperatures reported in supporting tables S2 and S4 refer to pH7 only. To what extent are the melting temperatures reduced at the lower pH postulated upon precipitation of acidified dew drops and have control experiments performed to evaluate this?

3) The experiment shown in Fig. 2 and Fig. S7, S8 for replication of sunY-derived RNA strands in AWI-based non-equilibrium environments were performed in the presence and absence of the template, but as far as I understand, always in the presence of full-length sunY. What would happen in the absence of full-length sunY but with added sunY-derived fragments is shown later in Figs. S14b, c and d. I think what is shown there is a very important result demonstrating that the fragmented version of the ribozyme is able to support ligation of the short substrates into the fragments that then assemble into the active ribozyme. This is even more compelling in the context of early life scenarios than ligation supported by the rather large full-length sunY. This should be much more focused on and discussed in the main manuscript.

4) I found it a bit difficult at times to assign the data to the particular system. Small schematics of the particular reactions/substrate-template complexes accompanying the original data (gels) in the figures would be helpful.

Reply to Reviewer Comments:

We have carefully considered and responded in detail to all points raised by the three reviewers and think that we were able to address all issues. Our responses as well as all changes made in the manuscript, are described in detail below. The comments are set in italic and highlighted in blue; our responses are shown in normal font. All major points (RX.X) as well as the respective answers (AX.X) are numbered to cross-reference more easily.

We would like to thank the three reviewers for the detailed comments and constructive and helpful feedback. We feel that the revisions improved our manuscript considerably.

Reviewer #1 (Remarks to the Author):

I think it's a well written paper, although it is hard to discern the major novelty points of this work. The experimental plan and rationale is very clear, has a place and the system is not needlessly complicated. The paper is well written, the narrative flows well and all data is presented in a convincing way. The figures are clear. The experiments are concise and streamlined.

We thank the reviewer for his/her assessment. Addressing the comments mentioned led to a more focused discussion on the novelty and impact of our experiments and helped to clarify some points for the more general scientific community.

RI.1:

I am unclear if the Authors actually make a dew-drop forming system? The descriptions suggests Authors just change the temperature and Mg concentration? Most of the time, the paper says "AWI conditions" or "AWI-like system," but sometimes it is "AWI system," which doesn't seem accurate unless they're remaking the rock-water interface from Fig 1A. It would be useful to clarify that and make it consistent through the text.

A1.1:

We apologize if this important point remained unclear to the reviewer. All experiments, except for the isothermal controls, were performed in the air-water interface (AWI) chambers illustrated in Fig. 1A, which were made specifically for this type of experiment. These systems mimic a partially submerged rock pore by sandwiching a Teflon cut-out between two sapphire windows. A more detailed description of how these chambers are created is shown in Figure S1 and the supplementary methods section 3.2. For the reactions performed under at air-water interfaces the Teflon chamber is filled with the reaction sample, carbon dioxide gas and exposed to a temperature gradient that stays constant throughout the experiment. The changes in temperature, pH and magnesium concentration are entirely driven by the dynamics that result from the temperature gradient across the whole system. In general, we refer to these self-built systems as “AWI-systems” or “AWI conditions”.

Nonetheless, we agree that introducing the abbreviation AWI as 'porous rock systems comprising heated air-water interfaces (AWIs)', can cause confusion as we are not including rock samples in our experiments. To avoid this confusion, we removed the abbreviation in line 53-55 and added the sentence: 'Of particular interest are porous rock systems comprising heated air-water interfaces (hereafter referred to as AWI-systems), which can be experimentally reproduced in a defined manner by microfabrication', in order to clarify that AWI is referring to the experimental setting. To avoid further confusion and for the sake of consistency, we removed the phrasing 'AWI conditions' or 'AWI(s)' and replaced it with 'AWI-system(s)' in figure captions and throughout the manuscript.

RI.2.:

How artificial having the G-leaving groups are? Are they truly prebiotically plausible? If not, still a really interesting and elegant paper, but it requires clarification. Either way, I think it would be very beneficial to the readers, placing it in the broader context of prebiotic evolution, to address how plausible their starting structures are pre-biotically. Perhaps also it would be useful to discuss what needs to be furthered to either make these G-activated RNAs or if there's any systems that don't need them.

AI.2:

We thank the referee for this comment. The advantage of using *sunY* as a ligase is that no dedicated chemical activation (such as triphosphates) is required for ligation since the ligation is based on the catalytic mechanism of group I introns. The reaction proceeds at nicked double-stranded structures that are bound to the ribozyme at the ligation junction via the G-binding pocket. As sketched in the now modified Figures 1B and S2, the resulting reaction proceeds by the nucleophilic attack of the terminal 3'-hydroxyl group on the phosphodiester bond between the proximal guanine and nucleotide after it, resulting in a G leaving group. In principle, also 5'-Gs with longer upstream sequence elements can act as leaving groups as well (which is the case during the natural splicing reaction) as long as the downstream sequence can bind to the ligation template and fulfils certain sequence constraints (see Doudna, Couture and Szostak, <https://www.science.org/doi/10.1126/science.1707185>). Thus, the ligated strands are as plausible or implausible as the prebiotic synthesis of RNA itself. Of course, the *sunY* system just serves as a proxy for prebiotic RNA-replication as it is not perfect e.g. in terms of fidelity.

To clarify this aspect, we slightly changed the sentence in lines 76-79 to '*Due to their robust activity and independence from prerequisite activation chemistries such as phosphate- or imidazole-based leaving groups, sunY derived ribozymes are an attractive model system for primitive enzymatic RNA-replication*'.

As reviewer 1 points out, other (model) systems for ribozyme-driven replication exist and we agree that it would benefit the reader to put this work in the context of those replication systems. We aim to adapt the setup to other ribozyme systems in the future. As temperature gradient and concentrations can be optimized for a given case, we added the following paragraph to the conclusion to stress the generality of this approach and the benefit it could offer to ribozymes with high salt requirements (lines 227-235):

“AWI-systems can be adapted to the requirements of other ribozyme systems by adapting temperatures or chamber geometries. Of particular interest are RNA polymerase ribozymes that catalyse templated primer extension reactions of equal sequence length to themselves. As previously mentioned, the problem of separating de novo RNA from its template is exacerbated by salts in solution. This is even more the case for RNA polymerase ribozymes that have been characterised under magnesium concentrations ranging from 50 – 200 mM⁶⁻⁸. It remains to be seen how these systems also benefit from the salt and pH oscillations to enhance the yields by the positive feedback of template release and by reduced metal ion- and pH-mediated degradation of catalytic RNAs.”

RI.3

While this is well written paper with elegant experiments, I am not sure if this is an advancement significant enough for a major publication. All elements of the system described here were presented before, and this work doesn't add a significant advancement to our understanding of prebiotic replication, hydrothermal cycling or ribozyme biochemistry.

A1.3:

We are surprised by this statement and respectfully but strongly disagree with it. To our knowledge, no one has been able to combine template-directed copying of RNA by a ribozyme with strand separation in a single reaction environment, thereby enabling both (+) and (-) strand replication. Furthermore, to our knowledge, no one has yet succeeded not only in copying the sequence of another ribozyme, but also in enabling it to dissociate from the template, fold, and become catalytically active in the same reaction environment where its own synthesis occurred. As reviewer 3 puts it: The paper ‘*contributes to the solution of a long-standing problem in the study of the origin of life, i.e. product inhibition during copying of a nucleic acid strand due to the high thermodynamic stability of the duplex formed*’.

Strand separation and re-annealing has long been a problem in the origin of life field. The problem has received much attention by both theoreticians and experimentalists, and many, including ourselves, have been working on possible solutions (see Introduction, lines >40). RNA duplexes of even 30-50 base-pairs are considered impossible to thermally denature under the conditions required for general RNA replication (Szostak, J.W. The eightfold path to non-enzymatic RNA replication. *J Syst Chem* **3**, 2 (2012). <https://doi.org/10.1186/1759-2208-3-2>). There is a trade-off between having conditions

under which duplexes can be melted and conditions, which allow RNA folding and therefore catalysis including ligation and polymerisation. We believe that heated air-water interfaces provide the advantages of both without suffering the consequences of efficiency.

Previous elegant solutions that circumvent this problem have been described such as the R3C ligase ribozyme (e.g. Paul and Joyce: 'A self-replicating ligase ribozyme'). However, these ribozymes cannot copy general sequence information - a prerequisite for open-ended evolution by natural selection.

We believe that the innovation of our approach was to leverage the dynamics of a non-equilibrium system to drive the one-pot synthesis of sense and antisense strands with very little sequence prerequisites, as well as the templated synthesis and release of a functional ribozyme itself.

In response to this and reviewer 2's comment (R2.1), we revised the abstract to emphasize the synthesis and functionality in a one pot reaction. We added 'enable sense and antisense RNA replication as well as template-dependent synthesis and catalysis of a functional ribozyme in a one-pot reaction' to the sentence in line 20-21. Additionally, we have reworked the discussion to elaborate on the implications of this work on future work.

Reviewer #2 (Remarks to the Author): Summary:

The manuscript Salditt et al. explores how the environment of a temperature gradient microchamber - as model system for a Hadean rock pore - can enhance, and facilitate processes related to the replication of catalytic RNAs. It combines three phenomena that were published previously (the temperature gradient induced formation of pH gradients that lead to destabilization of RNA duplexes, the temperature gradient induced localization of Mg²⁺ / RNA to the gas / water interface that lead to ribozyme activity under otherwise suboptimal Mg²⁺ concentrations, and the templated ligation of a hammerhead ribozyme by a group I intron (I'm not sure whether the latter is published- but it would not be a significant advance in itself)), and the novelty of this manuscript lies in the combined effect of these three phenomena. While there is likely also formation of a complete sunY ribozyme, this formation is not confirmed, which could be done either by reverse transcription and sequencing of several clones, or by showing catalytic activity of the sunY ribozyme. The concentration effect of RNA and Mg²⁺ at the gas/water interface allowed template-directed, ribozyme mediated ligation of three short oligonucleotides to occur at bulk Mg²⁺ concentrations that otherwise do not support efficient catalysis. Additionally, the effect of the CO₂ atmosphere generated a local pH decrease from dissolving CO₂ in dew droplets of the temperature gradient microchamber, which allowed strand separation of the products in detectable quantity. When the hammerhead ribozyme was formed by group I intron mediated, templated ligation, catalytically active hammerhead ribozyme was produced. I feel that the study may be publishable in Nature Communications if either replication is demonstrated as described below, or if the title, abstract, and text are toned down to correctly describe the findings.

We thank the reviewer for his/her assessment of our work and placing our work in the context of the current literature. We would like to add that RNA catalysis, and in particular template-directed RNA replication have never been performed in AWI interfaces exposed to a CO₂ atmosphere and, thus, the associated pH gradients forming in these systems. In addition, we would like to point out that we demonstrate full replication of two strands (the sunY fragments A and C), i.e. their de novo (+) and (-) strand synthesis. The ligation of the Hammerhead ribozyme was not carried out to demonstrate replication of its sequence, but that it is possible to combine *de novo* templated synthesis of a ribozyme with strand-release, folding and catalysis. We have made these differences now clearer in Figures 2 and 3 and their respective captions as well as in the main text (see A2.1).

Major points

R2.1:

The title and abstract is misleading, in two ways; the first is most important. First, there are no RNA replication cycles in this study, especially no complete ones. The manuscript shows that a sun Y ribozyme mediates the templated synthesis of a functional hammerhead ribozyme from three fragments. This is not replication, this is ligation / synthesis. For replication, at a minimum, both strands of the hammerhead ribozyme would have to be produced, and it would have to be shown that the two strands template each

other's synthesis. Alternatively, the synthesis of both strands of the subY ribozyme by the sunY ribozyme could be shown. So far, this is not contained in the manuscript because only one strand seems to be produced, and because the sequence of the ligation product is not confirmed. The latter could be done either by sequencing of the ligation products (at least 3 clones with correct sequence) or demonstrating catalytic activity of the product strand. The incorrect impression of replication in the title is enhanced by the word 'complete' in the title.

A2.1:

We thank the reviewer for the detailed analysis of the title and abstract. We agree that replication cycles involve the synthesis of both sense and antisense strands of a given sequence, and that these have to dissociate to template the next generation of ligation / synthesis reactions.

The referee writes: “(...) *the synthesis of both strands of the sunY ribozyme by the sunY ribozyme could be shown.*” We would like to stress that we indeed demonstrated exactly this kind of reaction for two individual fragments of the *sunY* ribozyme (data shown for fragment C123 in Figure 2b, c and FigureS8 for fragments A123 and C123): We initiated replication with unlabelled templates for each fragment (either sense or antisense), and observed in both cases the *de novo* formation of both, labelled sense and (!) antisense products. We apologize if these results may have not been adequately presented before. We have therefore revised Figure 2 to include illustrations of the reaction setup and conditions for each ligation reaction setup (also in response to R3.4). We have also made the figure captions clearer.

As already mentioned above, ligation of the hammerhead ribozyme was not performed to demonstrate replication of its sequence (the building blocks for the template had been omitted) but that it is possible to combine templated synthesis of a ribozyme with strand-release, folding and catalysis. To better guide the reader towards our findings, we modified the final sentence of the introduction, which now reads: “*In this work, we demonstrate how AWI systems allow ribozyme-catalysed RNA replication of sense and antisense strands followed by subsequent strand dissociation in a one-pot system. The combination of these reaction steps, which are otherwise mutually exclusive under isothermal conditions, enables the combined synthesis, release and folding of active ribozymes. Overall, these results infer that abundant geothermal microenvironments had the potential to support replication and thus evolution of early biological systems.*”

To avoid misunderstandings, we have changed the title of the manuscript now to: ‘**Ribozyme-mediated RNA synthesis and replication in a model Hadean microenvironment**’. Additionally, we have improved the clarity of the abstract. We hope this helps to avoid the misunderstanding that both phenomena of replication and functionality occur concomitantly.

Regarding the confirmation of the sequence identity of the ligation products: All syntheses of RNAs in this study is based on the template-dependent ligation of three substrate oligos. Therefore, cloning and sequencing of the products is, in our, opinion not required to verify the identity of the products for two reasons:

(1) We observe well defined intermediate and product bands but no major side-products in all our gels. Thus, mis-ligation events seem to be very rare and therefore negligible in our current system as they are typically guided by sequence-specific Watson-Crick base pairing of more than 7-10 nucleotides and are carried out in the absence of competing oligonucleotides with e.g. a similar sequence. We agree that fidelity of *sunY* would become an issue for shorter substrate oligos or mixtures of similar sequences (as explored in Doudna *et al.*, *Biochemistry* 1993, 32, 8, 2111–2115, <https://doi.org/10.1021/bi00059a032>). In such a case, we agree that sequencing of the products is absolutely necessary.

(2) In Figure 3 we ask the question: Do the sequences synthesized by *sunY* retain their function? We think we have shown nicely that they do, by synthesising the hammerhead ribozyme and quasi-simultaneously observing the cleavage of its cognate substrate. We agree that we have not shown both replication and activity concomitantly in the same experiment (replication for *sunY* fragments and activity assay for the hammerhead), but the results obtained suggest that the system is amicable to both phenomena.

R2.2:

*Second, while the temperature gradient chamber is a model system for a Hadean microenvironment it cannot be used to say that it *is* a Hadean microcompartment (in the absence of direct Hadean evidence). If there is no direct evidence that this microcompartment (with such a temperature gradient) existed in the Hadean then it needs to tone down such as 'Hadean rock crack model chamber' or get more specific such as '... in a CO₂-rich atmosphere and a temperature gradient microchamber as model system for Hadean rock pore'.*

A2.2:

We thank the reviewer for pointing out this mistake on our part. We agree that our setup is a model system for such presumably naturally occurring systems on Hadean Earth. Thus, we changed ‘Hadean microcompartment’ to ‘model Hadean microcompartment’. The full title now reads: **‘Ribozyme-mediated RNA synthesis and replication in a model Hadean microenvironment’**

R2.3:

Line 59 and 60-62: Since the presented system is based on several phenomena that are now well known in the wider scientific community, their background needs to be explained in more detail:

Following Reviewer 2’s remark, we included explanatory sections for each of the points as described below.

R2.3.1: *(i) The equilibrium that leads to the local low pH needs to be explained with a bit more detail, and more accurately. 'Carbonate' by definition is the (CO₃)²⁻ divalent anion - but this almost certainly does not exist under the conditions described. In contrast, this seems to be an equilibrium between dissolved CO₂ / carbonic acid and hydrogen carbonate, where the equilibrium is around pH 6.4.*

A2.3.1:

We thank the reviewer for pointing this out. Indeed, the dissolved CO₂ does not only lead to carbonate, but rather to an equilibrium of carbonic acid, bicarbonate and carbonate as shown in supplementary Figure S6 reproduced below. Following Reviewer 2's suggestion, we replaced 'carbonate' with 'an equilibrium between carbonic acid, bicarbonate and carbonate' at lines 64-66.

Concerning the remark about the equilibrium pH of 6.4, we agree that more background information is helpful to clarify the origin of pH 4. As per Henry's Law of the solubility of gases, the amount of gas dissolved in a liquid is directly proportional to the partial pressure of that gas in equilibrium with the liquid. This essentially means that higher CO₂ partial pressures lead to more dissolved CO₂, which in turn will further lower the pH of the dew droplets that form. This dependence was already described by Harold T. Byck in 'Effect of dissolved CO₂ on the pH of water' (<https://doi.org/10.1126/science.75.1938.224>). For the AWI-system, the influence of different partial pressures was studied in detail by Ianeselli *et al.* in 'Water cycles in a Hadean CO₂ atmosphere drive the evolution of long DNA', both experimentally and by finite-element simulations (<https://doi.org/10.1038/s41567-022-01516-z>):

Figure S1.3: Simulated Bjerrum plot in the dew. a) Bjerrum plot: molar fraction of the carbonate species as a function of pH at steady state (assumed to be steady after 600 seconds of simulation). CO₂ partial pressures in the range from 10 μ bar to 10 bar yielded a pH range from 7.0 to 3.3.

In the present study, the partial pressures in our setup was between 0.5 and 0.8 bar CO₂, resulting in a mean pH of approximately 4.0-4.3.

To take this more detailed consideration into account we inserted at line 65 ‘... *depending on the applied partial pressure of CO₂*’ The full sentence now reads: ‘In CO₂-rich AWIs, nucleic acids are exposed to periodic changes in Mg²⁺ concentration and pH level, the latter of which originates from an equilibrium of dissolved carbonate carbonic acid, bicarbonate and carbonate in dew droplets depending on the applied partial pressure of CO₂.’

R2.3.2: (ii) the ability of pH around 4 to partially melt duplexes, and

A2.3.2:

As Reviewer 3 also requested more detail concerning the melting ability at pH 4 compared to pH 7, we added columns with the respective melting temperatures of the formed duplexes at pH 3.6 and pH 4 for bulk salt concentrations and low salt concentrations in supplementary tables S2 and S4.

To the best of our knowledge, we are not aware of a conclusive study that sheds light on how or why pH destabilizes double-stranded nucleic acids. However, in the paper ‘pH-Driven RNA Strand Separation under Prebiotically Plausible Conditions’ (<https://doi.org/10.1021/acs.biochem.8b01080>), Sutherland and co-workers state: “As DNA is known to undergo reversible acid denaturation (presumably through protonation of GC base pairs and consequent formation of Hoogsteen base pairing), the melting temperature of RNA oligonucleotides is likely to be similarly influenced by the pH of the solution.” Thus, we included ‘presumably due to nucleobase protonation’ at line 67 and referred to the study mentioned above.

R2.3.3: (iii) the accumulation of RNA and Mg²⁺ at the water/gas interface need to be explained, each with perhaps one sentence.

A2.3.3:

The accumulation RNA and Mg²⁺ both result from the same cause. The heated gas-water interface causes continuous evaporation-recondensation cycles, with molecules accumulating at the hot side due to a superposition of the following flows: the capillary flow at the meniscus, the diffusion of water vapor between the interface and the gas bubble, the convection of water and (to a lower degree) the Marangoni flow along the interface (Morasch *et al.* ‘Heated gas bubbles enrich, crystallize, dry, phosphorylate and encapsulate prebiotic molecules’ <https://doi.org/10.1038/s41557-019-0299-5>). The difference in accumulation strength results from the interplay of the different strengths of back-diffusion and capillary flow.

To elucidate this point, we added ‘*This accumulation is the result of the capillary flow created by water evaporation at warm side of the air-water interface. The effect of water convection and Marangoni flow at the interface is minimal, but contribute if molecular assemblies grow to the size of tens of micrometers.*’ in lines 69-72 of the introduction

Minor points

Line 15: 'stored on' -> 'stored in'

Corrected.

Line 16-17: There is at least one additional hurdle why 'experimental systems for sustained RNA-dependent RNA-replication are difficult to realize': Only very few systems are available, and all of them are quite limited (I know of four: van Kiedrowsky's replicating hexanucleotide, Joyce's replicating R3C ligase ribozyme, Joyce's cross-replicator, and replicating group I introns, one of which that is the model in this study). I suggest either (i) inserting a 'partially'/ 'in part' such that it reads '... are difficult to realize, in part due to the high ...' or (ii) include additional hurdles such as the sparsity of designed and/or selected RNA systems described above.

We agree that there are more difficulties that need to be addressed and inserted ‘in part’ in the abstract as the word count of the abstract is limited, and the stated hurdles are the ones being addressed in this work.

Line 20-21: The abstract should include the information where these salt and pH oscillations come from (since there was no Hadean experimenter who added salt / water and acid / base). For example, "wet / dry cycling that resulted in oscillations of salt concentrations ...".

We agree that it is a significant information to include in the abstract and inserted '*resulting from precipitation of acidified dew droplets*' to the sentence '*Both reactions are driven by autonomous oscillations in salt concentrations and pH, resulting from precipitation of acidified dew droplets, which transiently destabilise RNA duplexes.*' (lines 21-23)

Line 33: Not all ribozymes require high magnesium concentrations. For example, group I introns are fine with 5 mM Mg²⁺, and some self-cleaving ribozymes can even work completely without Mg²⁺. Please tone down. Alternatively, if specific ribozymes are envisioned (see my comment to lines 16/17), please list them so that the reader knows what systems are discussed (without looking at the references).

We acknowledge that not all ribozymes require high salt concentrations for catalysis. Consequently, we have toned down the statement in the introduction to say that RNA duplex stability is '*...further enhanced by salts in solution which are required to fold the ribozymes and often for catalysis, ...*' (lines 34-35). To return to the point of which ribozymes we had envisioned could benefit from the AWI-system, we also refer to ribozyme polymerases specifically in the discussion and specify the magnesium concentration under which they have been characterised. (lines 227-235).

Line 77 and 81: ". concentrations..." -> "... bulk concentrations ..." because the concentration at the gas/water interface is significantly higher, which allows catalysis to occur.

We inserted '*bulk*' in both sentences. (lines 86 and 89).

Line 80: The word 'drive' is incorrect here because the driving force of the reaction is the leaving of the 5'-guanosine leaving groups. I suggest using 'mediate' or 'facilitate'.

We re-wrote the sentence (now lines 91-93). It now reads:

"These findings confirmed that the local up-concentration of solutes at the warm side of the AWI interface was sufficient to allow sunY-dependent RNA ligation at bulk magnesium concentrations considerably lower than those required under isothermal conditions."

Fig. 3: The strand of HH-min and the three fragments HH1-3 should be explicitly labeled in the figure since HH-min and HH1-3 have the same color. It becomes clear after making sense of the figure - but it should be clear without first understanding the figure.

We agree that schemes would benefit from this clarification and thank the reviewer for the suggestion. The respective labels have been added accordingly to the reaction schemes in Figures 2B, 2C and 3A.

Reviewer #3 (Remarks to the Author):

The manuscript by Salditt et al. reports an experimental scenario of sustained RNA replication at heated air-water interfaces in a CO₂-rich atmosphere. It is a very nice work that contributes to the solution of a long-standing problem in the study of the origin of life, i.e. product inhibition during copying of a nucleic acid strand due to the high thermodynamic stability of the duplex formed. Using a sophisticated experimental setup, the authors show that oscillations in salt concentration and pH drive the destabilization of the duplex, allowing iterative cycles of copying and replication of functional RNAs. The paper is well written, the data presented are reliable, and the conclusions drawn are reasonable and supported by the work.

We thank reviewer 3 for his/her assessment and understanding of our study. The comments helped clarify the manuscript and refine the outlook regarding the implications the origin of life field.

I have only a few minor comments that should be considered before publication:

R3.1:

SunY was used as an RNA ligase. Although sunY is shown schematically and also in sequence (Fig. S2), it would be helpful to show how the template-substrate complex interacts with the ribozyme to become ligated. Since not all readers are familiar with the exact mechanism of sunY, it might also be useful to elaborate on the catalytic center with the ligation site and the position of the guanosine residue that acts as the leaving group in this process.

A3.1:

We thank the referee for this thoughtful comment. We extended Figure 1B with a simplified illustration of the ribozyme-substrate complex and elaborated the figure caption with a more detailed description of the catalytic mechanism. Additionally, we highlighted the G binding pocket in the *sunY* sequence of Figure S2.

These changes are based on the recently published structure of a group 1 intron determined by Cryo-EM (see Su, Z., Zhang, K., Kappel, K. *et al.* 2021, ‘Cryo-EM structures of full-length *Tetrahymena* ribozyme at 3.1 Å resolution’, <https://doi.org/10.1038/s41586-021-03803-w>). The article elucidates how a large conformational change is critical for substrate docking and catalysis and offers detailed insights into the catalytic center of the intron.

Taking into consideration the complexity of structural features of group 1 introns presented in the above paper, we decided to exclude excessive details about the complex underlying structural details including the active site but instead refer to the article.

R3.2:

On page 4, beginning at line 82, it states, " We hypothesized that the periodic local pH decreases, resulting from precipitation of acidified dew droplets (Figure S6), could lead to a transient decrease in RNA melting temperatures thereby promoting the release of ligation products from their template and allowing intramolecular folding into functional RNAs." Has the exact pH at the air-water interface where replication occurs been determined? The melting temperatures reported in supporting tables S2 and S4 refer to pH7 only. To what extent are the melting temperatures reduced at the lower pH postulated upon precipitation of acidified dew drops and have control experiments performed to evaluate this?

A3.2:

We agree that this is an important point that benefits from further clarification. The dependence of the pH of the dew drops on CO₂ pressure in the AWI system was characterized in detail in 'Water cycles in a Hadean CO₂ atmosphere drive the evolution of long DNA' by Ianeselli *et al* using a pH sensitive dye (<https://doi.org/10.1038/s41567-022-01516-z>). The pH of the dew droplets for a CO₂ pressure of 1 bar on average was pH 4.0 (± 0.4), reaching transient pH values of even lower value. Additionally, Ianeselli *et al.* systematically probed the melting temperatures of nucleic acids with respect to length, GC content, mono- and divalent ion concentrations and pH. The melting temperatures of Supplementary Tables S2 and S4 are based on this data.

The different parameters that lead to the change in melting temperatures of RNA in the AWI-system are diverse and difficult to disentangle. As a result, the exact mechanism of strand separation is not trivial to infer. First, the acidic pH leads to a drop of the melting temperature, which supposedly can protonate the nitrogen atoms of the bases, leading to repulsive electrostatic forces that oppose hydrogen bonding. The average of pH 4 might not be enough to melt all of the described strands. However, the dew droplets do not have a uniform pH value but rather the pH

varies in different regions within the droplets, and can reach transient pH values ~ 3 . In addition to the low pH, the dew droplets are also low in salt (~ 0.25 mM see Ianeselli *et al.*), further decreasing the melting temperature of RNA/DNA duplexes.

To address this comment, we now included additional columns in Supplementary Tables S2 and S4 with estimates for the range of melting temperatures in the dew droplets. The tables now contain the melting temperatures for the mean (pH 4) and lower boundary (pH 3.6) of pH values with bulk salt concentrations, as well as the melting temperatures for both pH values with monovalent and divalent ion concentrations of 1 mM and 0.2 mM, respectively.

R3.3:

The experiment shown in Fig. 2 and Fig. S7, S8 for replication of sunY-derived RNA strands in AWI-based non-equilibrium environments were performed in the presence and absence of the template, but as far as I understand, always in the presence of full-length sunY. What would happen in the absence of full-length sunY but with added sunY-derived fragments is shown later in Figs. S14b, c and d. I think what is shown there is a very important result demonstrating that the fragmented version of the ribozyme is able to support ligation of the short substrates into the fragments that then assemble into the active ribozyme. This is even more compelling in the context of early life scenarios than ligation supported by the rather large full-length sunY. This should be much more focused on and discussed in the main manuscript.

A3.3:

We thank the reviewer for acknowledging the importance of this result. We agree that fragmented ribozymes that assemble intermolecularly into an active form are an interesting early life scenario. This also benefits the AWI system by lowering the threshold to achieve separation, as well as potentially pave way to assemble larger, more complex ribozymes with interesting functions. While the *sunY* retains activity in its fragmented form, it is significantly lower and thus produces less product compared to the full-length ribozyme. We attempted to synthesize the different fragments independently using fragmented *sunY* as the only catalyst, but the reaction time had to be adapted to much longer times (18 h instead of 2-4 h) to observe low amounts of product forming. We are aiming to eventually having the ribozyme synthesize all fragments of itself simultaneously in autocatalytic mode. Nevertheless, the autocatalytic replication of the fragmented *sunY* still requires much optimization, and we will proceed with this avenue of research in the near future.

However, we agree that this still poses an important result and expanded on this point in the manuscript. We omitted the section on the fragmented version in line 159 and instead inserted a more detailed discussion at lines 210 – 226, where we now elaborate on the difference of the ligation by the fragmented *sunY* of the hammerhead and the individual strands of fragmented *sunY*. We also hint at possible autocatalytic self-replication of fragmented *sunY* ribozyme, and go a step further and discuss the possibility of AWI-systems to host polymerase ribozymes, which also have the potential for self-replication. The inserted text is shown below:

“In its current form, the AWI-system is well suited to ligating RNA strands between 20 and 75 nt from much shorter precursor oligonucleotides, which have the potential to assemble into more complex ribozymes. Typically, fragmented ribozyme variants are in most cases less active than their full-length counterparts presumably to their lower stability and folding defects. In line with this, the fragmented sunY variant formed by noncovalent assembly of A123, B123, and C123, unlike the full-length variant, did not show detectable ligation of HH-min under standard AWI or isothermal conditions (Figure S14A).

Interestingly, however, we found that lowering the temperature on the warm side of the AWI system from 45 °C to 40 °C allowed the fragmented sunY ribozyme to synthesize HH-min, illustrating that milder conditions could compensate to some extent for the lower activity of the split variant (Figure S14A). In a first attempt to explore the autocatalytic potential of the system, we demonstrated the synthesis of all three sunY fragments (A123, B123, and C123) directly by the fragmented sunY, although the yields were considerably lower than for ligations catalysed by the full-length ribozyme (Figure S14B-D). Further optimization will be required to improve reaction yields, but the results provide an optimistic outlook on the capability of the system to undergo self-replication from a pool of shorter oligonucleotide fragments.”

R3.4:

I found it a bit difficult at times to assign the data to the particular system. Small schematics of the particular reactions/substrate-template complexes accompanying the original data (gels) in the figures would be helpful.

A3.4:

We agree that the figures benefit from this clarification. We elaborated on the reaction schematic to Figure 2 that describe the reaction going on, adapting it to each condition tested. To avoid overcrowding the original data with text and labels, we instead added labels to the strands of the schemes in Figures 2B, 2C and 3A.

REVIEWERS' COMMENTS

Reviewer #2 (Remarks to the Author):

The authors have satisfactorily addressed all my questions. I support the publication of the revised manuscript.

Reviewer #3 (Remarks to the Author):

The revised version of the manuscript reads very well. I find all my comments adequately taken into account and, as far as I can tell, the comments of the other reviewers have also been seriously considered in the revision. I liked the manuscript already at the first review, the clarification of the still open questions has improved it even further.